# Elementary School First Graders' Acquisition of Productive L2 French Grammar in Regular and CLIL Programs

**Patricia Uhl** *, **Anja K. Steinlen** and **Thorsten Piske**

Chair of Foreign Language Education, Friedrich-Alexander-Universität Erlangen-Nürnberg, 91054 Erlangen, Germany; anja.steinlen@fau.de (A.K.S.); thorsten.piske@fau.de (T.P.)
* Correspondence: patricia.uhl@fau.de

**Abstract:** This study presents productive L2 French grammar data obtained at the end of grade 1 from 186 elementary school children learning French in bilingual (CLIL) or in regular school programs in Germany. The children completed a picture description task to assess their productive oral L2 French grammar skills and two standardized cognitive tests on nonverbal intelligence and sustained attention. The results did not indicate any significant effects of the cognitive tests or of child-internal variables (in this case gender, language background and educational background). However, children in the regular French program unexpectedly outperformed their peers in the bilingual French program. Classroom observations and information provided by teachers suggest that this finding may, at least in part, be due to the fact that in grade 1 there were only minor differences between the two programs in terms of L2 exposure time and teaching methodology.

**Keywords:** productive grammar; attention; nonverbal intelligence; L2 French; multilingual background

## 1. Introduction

In second language (L2) acquisition research, various factors have been shown to affect the acquisition of L2 grammar, and these have often been subdivided into child-internal factors (e.g., language aptitude, motivation and, of course, individual cognitive abilities, such as attention or intelligence) and child-external factors (i.e., L2 input quality and quantity (e.g., Kersten et al. 2021; Paradis 2011; Unsworth et al. 2015)). The purpose of this article is to examine the influence of two of these factors in instructed second language acquisition, i.e., L2 input quantity and young learners' cognitive abilities. We will present data on children in grade 1 of elementary school, who had just started to learn French as a L2.[1] The key question addressed in this paper is how first graders' productive L2 French grammar is affected by their individual cognitive abilities (operationalized as nonverbal intelligence and attention) on the one hand, and by L2 input quantity (operationalized as regular vs. bilingual teaching programs) on the other hand.

In Germany, a L2 is first introduced in elementary school (with two lessons per week), and the target language is usually English. French as a regular subject is offered in only 3.7% of all elementary schools in Germany (Eurostat 2016). Most of these schools are located in federal states next to the French border, e.g., Baden-Württemberg, Rhineland-Palatinate and Saarland (KMK—Kulturministerkonferenz 2013). In addition, about 2% of all elementary schools in Germany offer a more intensive L2 program, namely one in which content subjects such as science, math or music are, at least partially, taught in a L2, and 13% of these bilingual schools offer a program with the target language French (FMKS, Verein für frühe Mehrsprachigkeit in Kindertagesstatten und Schulen 2014). In Europe the term CLIL[2] (Content and Language Integrated Learning) is often used to refer to such bilingual programs (e.g., Eurydice 2006). CLIL is an umbrella term that may refer to very different types of bilingual programs ranging from low-intensity programs with bilingual 'modules' or 'projects' in selected school subjects to high-intensity programs,

such as immersion programs in which at least 50% of the curriculum is taught through the medium of a foreign language (e.g., Kersten 2019).

In 2017/2018, the federal state of Bavaria initiated a pilot project in eleven public elementary schools to enable students to learn French as a L2 before they enter secondary school (Uhl et al. 2020). In these schools, French is either taught as a regular subject (on a voluntary basis) or it is used as the language of instruction in a CLIL program from grade 1 onwards (see Section 2.1 for more information). So far, French has not been studied in terms of productive grammar acquisition in such a context.

### 1.1. L2 Grammar Acquisition

In general, the acquisition of grammar may be defined as the acquisition of language rules and structures and the ability to use them in a communicative context, which applies to first (L1) and second language learning contexts alike (e.g., Nassaji 2017, p. 205). A very large number of studies are concerned with productive L2 grammar acquisition in general, in particular with its systematicity, as shown by morpheme order studies (see e.g., Ellis 2008 for a review) or by studies conducted in the context of Processability Theory (PT), a psycholinguistic theory of L2 grammar acquisition which explicitly predicts the order in which L2 learners learn to process different morpho-syntactic phenomena (e.g., Buyl and Housen 2015; Pienemann 2005).[3] The L2 examined in most of these studies is English. However, studies on L2 French have reported that children's L1 and L2 French grammar (e.g., morpho-syntax or phrasal structure) develop in developmental sequences similar to those that have been reported for English (e.g., Meisel 2009; Ziegler 2006).

Most studies examining productive L2 French grammar in CLIL settings have been conducted in Canada in intensive bilingual immersion (IM) contexts. These studies have often focused on the effects of different types of L2 grammar instruction (e.g., Lyster 2007). However, studies of very young learners' L2 French morpho-syntactic development are scarce: Harley (1984) reported that first graders in early French immersion programs make use of general-purpose verbs (e.g., *aller* ('to go'), *faire* ('to do')), which they also use for situations where verbs with more specific meanings are not available. They also use singular and plural noun phrases (*le garçon* ('the boy'), *les garçons* ('the boys')), some students even produce tense distinctions (e.g., past, present, future), although not consistently. Plural verb forms are not produced, and the word order in French sentences are generally similar to L1 English. According to Genesee (1987), any students' L2 production grammar is shaped in important ways by: (1) their L1 grammar system, (2) the communicative demands made on them in the bilingual classroom, and (3) the type of native speaker models they are exposed to. However, these aspects of grammar learning have not been studied for less intensive L2 CLIL French programs yet.

### 1.2. Cognition and L2 Grammar

In Germany and elsewhere, CLIL programs are often considered to be 'elitist' because they are often attended by students with particular personal, intellectual, or familial characteristics. When schools preselect their students, student-selection factors often include age-appropriate knowledge of the L1, the ability to concentrate, perseverance, commitment and/or communication abilities (e.g., Kersten et al. 2010), which may contribute to CLIL students outperforming students in regular programs on any language test, but also in receptive L2 grammar (e.g., Steinlen 2021). Previous studies examining language learning outcomes in CLIL have shown that students' foreign language test scores are higher than those of their non-CLIL counterparts: young learners in bilingual/CLIL programs in Europe have been found to generally acquire a wider vocabulary range, to show greater fluency and better receptive skills, as well as a higher degree of motivation and confidence to speak the L2 than students receiving formal L2 instruction (e.g., Eurydice 2006; Gebauer et al. 2013; Steinlen 2021; Wode 2009; Zaunbauer et al. 2012). However, CLIL students' productive skills do not always match up to their receptive skills and less gains have been reported in pronunciation and grammatical accuracy (see e.g., Dalton-Puffer 2011;

Pérez-Cañado 2012; Wesche 2002 for overviews). Furthermore, L2 intensity, i.e., the actual amount of students' L2 exposure and use, in school programs could be more important than the difference between regular L2 lessons and bilingual lessons in affecting learners' L2 competences considerably (Steinlen 2021).

The individual cognitive abilities of each learner play a very important role in the acquisition of languages. For L1 grammar, these include, among others, working memory, i.e., the active memory system that is responsible for the temporary maintenance and simultaneous processing of (grammatical) information (Bayliss et al. 2005), short-term phonological memory, i.e., "the ability to temporarily maintain speech-related information through a combination of passive and active mechanisms" (Fietz 2016, p. 855), grammatical awareness, i.e., the ability to identify and correct ungrammatical sentences, as well as to understand and use grammatical metalanguage (Andrews 1999), and verbal intelligence (e.g., Talli and Stavrakaki 2020). For L2 instructed learning, verbal short-term memory seems to be more strongly related to vocabulary learning, whereas verbal working memory is related to grammar learning (e.g., Verhagen and Leseman 2016). However, this study examines the relationship between L2 grammar and nonverbal intelligence on the one hand and between L2 grammar and attention on the other hand, and there are only a few studies on these relationships (see below).

### 1.2.1. Nonverbal Intelligence and L2 Grammar

Because IQ tests generally require the participants in a study to know the language well in order to be able to answer the questions in the test, nonverbal intelligence tests have often been employed for children and adult immigrants (Kuschner 2013). Nonverbal intelligence, as a cognitive skill, relates to the manipulation of visual information or problem solving on a visual level and may vary in the amount of internalized, abstract, or conceptual reasoning, as well as with regard to the motor skills required to complete a task. Different instruments can be used to test this particular cognitive skill. A frequently used measure are the Coloured Progressive Matrices (CPM, Raven 1976), where the participants' task is to find a missing piece in a geometrical pattern, using one of six possible alternatives. So far, the results from several studies comparing children's performance in Progressive Matrices tests in CLIL and regular programs are inconclusive: some studies have found significant between-group differences (see e.g., Zaunbauer and Möller 2007), while others have not (e.g., Lambert and Tucker 1972; Yadollahi et al. 2020). Preselection effects and/or positive long-term cognitive effects of intensive L2 exposure, especially regarding executive control (e.g., Bialystok and Craik 2022), may be the cause of significant between-group differences, while balanced results can be interpreted as the effects of equal opportunities in the selection procedure for bilingual programs and/or the lack of cognitive effects, for example, due to infrequent L2 exposure or the interaction with other decisive factors, such as socioeconomic background or motivation. In some studies, nonverbal intelligence has been used as a covariate when comparing students' language proficiency in mainstream and bilingual programs, with the aim to control cognitive skills when examining the effects of different school programs (e.g., Steinlen 2021). Based on the findings reported above a correlation is expected between the measures of nonverbal intelligence and grammar, which has been found in a number of L1 studies (e.g., Dąbrowska et al. 2020).

According to Kristiansen (1990), nonverbal intelligence and grammar tests share some characteristics: For example, in order to master a language, one must be able to produce grammatically correct sentences. Thus, a prerequisite for understanding and using grammatical structures correctly includes the ability to make inferences and to analyze tasks, and these are exactly the same abilities as those needed for most nonverbal intelligence tests. However, only very few studies have examined the relationship between nonverbal intelligence and L2 grammar: for adults, Brooks and Kempe (2013) reported nonverbal intelligence to predict L2 production of Russian gender agreement and case marking, concluding that it strongly modulates success in explicit inductive L2 grammar rule discovery. Sun (2015) examined how 41 Chinese preschoolers receiving two hours of

English instruction per week at an English school in China performed in an L2 English receptive grammar test. She found that analytical reasoning ability, as measured by the CPM, significantly predicted English receptive grammar outcomes. Sun (2015) pointed out that in contexts where L2 input and output are scarce, analytical reasoning ability might emerge as a more significant factor than memory in dealing with sentences, because it may help children to better organize the intensive and complicated information included in sentences. The effects of nonverbal intelligence on L2 French grammar in CLIL vs. regular L2 programs have not been examined yet.

### 1.2.2. Attention and L2 Grammar

Another cognitive variable examined in this study is attention. It is characterized as the behavioral and cognitive process of selectively concentrating on a discrete aspect of information (either subjective or objective), while ignoring other perceivable information. Attention has also been described as the allocation of limited cognitive processing resources, and it is a very basic function that is often a precursor to other neurological/cognitive functions (Anderson 2004, p. 519). Although the term attention has been defined in many different ways in the literature, according to Smith and Kosslyn (2006), there is broad consensus that attention involves selecting some information for further processing and inhibiting other information from receiving further processing. Especially in the German context, the terms concentration and attention are often used synonymously, although concentration is usually defined as a measure for the intensity and duration of attention (e.g., Westhoff and Hagemeister 2020). Concentration may thus be characterized as sustained attention, i.e., the ability to maintain a consistent behavioral response during continuous and repetitive activity. Other components, which are usually part of any model on attention (e.g., focused, selective, alternating and divided attention (e.g., Sohlberg and Mateer 1989)), do not constitute the focus of the present study.

In the school context, sustained attention is a key element because it determines academic success (e.g., Rhoades et al. 2011). However, children's attention is subject to age. For example, the average attention span for a preschooler is usually less than fifteen minutes, and for elementary school children language teachers are advised to limit activities to ten minutes (e.g., Harmer 2005). Child-internal factors may also impede their attention and, in the long run, their success in L2 learning in school: about 5–10% of school-aged children experience learning and social functioning problems caused by attention deficit (hyperactivity) disorder, corresponding to 1–2 students per class (e.g., Janicka 2015). The only study we are aware of which has assessed attention/concentration in early L2 learning is a study by Zaunbauer and Möller (2007), who compared first graders in immersion and regular English programs, which employed a test on attention/concentration (Möhling and Raatz 1974). They did not find any significant between-group differences, but they did find age-appropriate mean scores for both groups. Unfortunately, they did not specify whether the preselection processes or cognitive effects of the teaching method could explain these results. Studies on the relationship between sustained attention and grammar learning are also rare: West et al. (2021) examined 112 seven-year-old children using the English receptive grammar test TROG-2 (Bishop 2003), which measures the comprehension of 20 constructs four times each by using different test stimuli. They also measured attention during a serial reaction time task and found attention to be a predictor for receptive grammar. Whether this also holds true for productive L2 French grammar is examined in this study.

Beyond cognitive skills (e.g., Kersten et al. 2021) other learner factors, such as the language program, the students' gender, their language background and their educational background can serve as predictors of the students' L2 performance (e.g., Böhme et al. 2016; Carr and Pauwels 2006; Gottburgsen and Gross 2012; Paradis 2011). In particular, the educational background is important, due to possible preselection effects as described above. For this reason, these background factors are also taken into consideration in this study.

### 1.3. Research Questions

Based on the literature review, the following research questions pertaining to first graders' foreign language grammatical skills will be addressed:

1. Do children in bilingual and regular French programs differ with regard to their foreign language grammar skills after the first year of exposure, due to differences in L2 intensity?
2. Do the children in these programs differ with respect to their cognitive backgrounds (i.e., attention and nonverbal intelligence), due to possible preselection effects?
3. Do variables such as the language program, cognitive skills, the students' gender, their language background or their educational background serve as significant predictors of first graders' productive L2 French grammar performance?

## 2. Method

The following sections provide information on the pilot project, which implements French in eleven public elementary schools in Bavaria, either as a regular (but voluntary) L2 subject (i.e., French-as-a-subject, abbreviated as FL2) program or in bilingual (CLIL) programs, which all start in grade 1.

### 2.1. The Context

The current study draws on data obtained from eight of the eleven schools participating in the Bavarian pilot project. Four of these eight schools offer both CLIL and a regular French as a foreign language program (in which French is taught as a subject in two 45 min lessons per week). Two of the eight schools exclusively offer CLIL and the remaining two only regular French (i.e., FL2) classes. On the whole, six CLIL groups and nine regular L2 classes were involved in the study.

In the bilingual CLIL programs, 25–30% of the CLIL teaching time is conducted in French (in the subjects math, science, art, music and/or physical education). In each of the CLIL subjects the teachers themselves can decide which parts of the curriculum appear to be appropriate to be taught in French, and approximately one third of the teaching time in each of these subjects is carried out in French. After four years the students are expected to attain at least level A1 in French, which is a comparatively small competency goal. With regard to productive L2 French grammar skills, level A1 means that the learners show "only limited control of a few simple grammatical structures and sentence patterns in a learnt repertoire" (Council of Europe 2018, p. 133). Within this context, this means that the students should be able to use the first person singular and the third person singular and the plural of some frequent verbs, such as *être* ('to be'), *avoir* ('to have'), *faire* ('to do'), *venir* ('to come'), *s'appeler* ('to be called') or *parler* ('to speak'/'to talk') in *présent de l'indicatif*. Beyond that, the learners are also able to use some personal pronouns, such as *je* ('I'), *il/elle* ('he/she'), *ils/elles* ('they'), *moi* ('me'), *lui/elle* ('him/her'), some prepositions for location *être à/dans/au/en* ('to be at/in/in the/on'), as well as *présentateurs c'est* ('this is') and *voilà* ('here is/are'). Furthermore, the children can distinguish between the definite (*le/la/les* ('the')) and the indefinite articles (*un/une/des* ('a/some')), can use some cardinal numbers (*premier* ('first'), *deuxième* ('second')), begin to match the adjective with the noun (*accord*) and can use some demonstrative (*ce, cette, cet, ces* ('this/these')) and possessive determiners (*mon/ma, ton/ta, son/sa* ('my/your/his/her')), adverbs of quantity (*un peu* ('a little'), *beaucoup* ('a lot')), the partitive article (*du/de la/des* ('Ø/some') and some conjunctions (*et* ('and'), *ou* ('or') (Chauvet 2008, p. 31). Unlike the regular program in which French is taught as a subject (e.g., FL2 classes), the focus of the French CLIL lessons is on the acquisition of subject content, based on the respective curricula (Bayerisches Staatsministerium für Unterricht und Kultus 2014, 2021). Thus, grammatical structures are not taught explicitly, but are embedded in the context of teaching a particular topic in a particular subject. In both programs, the teachers' educational degrees meet high criteria: all of them studied French at university and obtained at least level C1.[4] Specific textbooks are neither used in the

regular nor in the bilingual program, because textbooks for French lessons in elementary school have not yet been approved in Bavaria.

Since the beginning of the project, a competence framework has been designed for the FL2 classes to establish curricular standards (Schwanke and Uhl 2020, p. 50). In the first and second year of learning, grammar teaching focuses on French morpho-syntactic structures, in particular declarative and interrogative sentences, the conjugation system of frequent verbs in *présent de l'indicatif* (e.g., *être* ('to be'), *avoir* ('to have'), *faire* ('to do')), personal pronouns, and in-/definite articles. During these first two years of learning, students also learn several frequent nouns in their singular and plural forms (e.g., *un ami* ('a friend'), *les amis* ('the friends')) and their combinations with numerals, prepositions and adjectives (e.g., *cinq pommes* ('five apples'), *une chemise bleue* ('a blue shirt')).

First graders in Germany do not usually receive any explicit grammar instruction in the early foreign language classroom due to the students' age, as well as curricular restrictions, and this applies to regular and bilingual programs alike. Nevertheless, sometimes teachers, particularly in bilingual programs, feel the need to address a particular grammatical phenomenon because either the students explicitly demand an explanation or specific grammatical activities seem to help the children to better understand a grammatical structure (e.g., Doughty and Williams 2008; Elsner and Keßler 2001, pp. 177–78). In such cases, teachers often turn to awareness raising activities which direct learners' attention to language form or "focus on form," or to metalinguistic explanations used, for example, when meaningful input is provided through stories, songs etc. (see Robinson 2017 for a review), and both approaches have been found to be beneficial even for young L2 learners.

*2.2. Test Materials*

(1) <u>Nonverbal intelligence</u>: Cognitive background data were assessed because test scores pertaining to linguistic or academic achievements may be affected by children's cognitive skills (e.g., Steinlen 2021). Both cognitive tests were employed as group tests. The Coloured Progressive Matrices (CPM, Raven 1976) is a measure of nonverbal intelligence, and the children's task was to complete an incomplete geometrical pattern using one of six possible alternatives. The test consisted of 36 items, which were presented in 3 sets of 12, in increasing order of difficulty within each set. 20 to 30 min were allocated for the test. The publishers of the German version of the test reported the internal consistencies to lie between $r = 0.80$ and $r = 0.90$ and the CPM to be a good indicator for Spearman's g-factor, which also yielded satisfying correlations with school performance tests (Bulheller and Häcker 2010);

(2) <u>Attention/concentration</u>: For grade 1, the ability to concentrate was assessed by means of the *Konzentrationstest für das erste Schuljahr* (KT1, Möhling and Raatz 1974). This timed paper-and-pencil test requires students to mark all the pears located between apples, i.e., to distinguish between similar visual stimuli within a time span of one minute. As a measure of the ability to concentrate, the number of marked items minus the omissions and errors was used. A maximum of 48 points could be reached. Möhling and Raatz (1974) reported a reliability value of $r = 0.75$ and a split-half correlation coefficient of $r = 0.82$;

(3) <u>Productive L2 French grammar test</u>: Because standardized tests for young beginner L2 French learners are not available, an individual test for free monologue-descriptive speaking (FREMODS) was designed, which is based on the work of Diehr and Frisch (2008). After some introductory questions (e.g., *Ça va ?* ('How are you?')), each student was asked to describe a picture (*Qu'est-ce que tu vois sur l'image ?* ('What do you see in the picture?)), which depicted people's activities in a park. Two minutes were allocated for this French speaking task, which was tape recorded. The children's utterances were later assessed with respect to lexis, morpho-syntax, pronunciation and fluency. For morpho-syntax (FREMODS-MS), 0 points were assigned when the student said nothing or only words or sentences in languages other than French (e.g., German or English). Moreover, 1 point was assigned when simple words without articles or numerals were mentioned, 2 points were given when nouns with articles and/or numerals were produced, and 3 points when the

student formed at least one whole and correct sentence (e.g., *Le soleil brille.* ('The sun is shining')). Thus, a maximum of 3 points could be attained for morpho-syntax. For the present sample, Cronbach's $\alpha$ was 0.884, for the common score for lexis, morpho-syntax, pronunciation and fluency and, altogether, a maximum of 10 points could be reached in these four fields;

(4) Parent questionnaire: A questionnaire was distributed which required the parents to indicate how many and which languages their child spoke at home with different family members. They were also asked to specify how much time their child spent using each of the languages s/he knew. In addition to this, the parents reported on their highest school-leaving qualifications, which were codified afterwards.[5] The mean values of the parents' school-leaving qualifications were used as an indication of the educational background.

*2.3. Participants*

Overall, 73 first graders in the regular FL2 program (51% female, 37% multilingual) participated in the speaking test, which assessed productive L2 French grammar. The participants' average age was 7.3 years (standard deviation, SD: 4.6 months). Pupils with French as L1 were excluded from the data set from the very beginning. The multilingual first graders in the regular FL2 program spoke a large variety of heritage languages, i.e., Russian, Polish, Spanish, Arabic, Albanian, English, Hindi, Greek, Swedish, Romanian, Kurdish, Bosnian, Croatian, Punjabi, Amharic, Pakistani, Ewe, Turkish and Malayalam (languages ordered according to their frequency in the sample).

In the CLIL program, 113 first graders (51% female, 42% multilingual) at an age of 7.25 years (SD: 4.4 months) also participated in the study. The heritage languages spoken by the first graders in the CLIL program were Russian, Arabic, Turkish, Spanish, English, Vietnamese, Albanian, Croatian, Punjabi, Azerbaijani, Amharic, Korean, Persian, Italian, Greek, Georgian, Bosnian and Hindi. The high proportion of multilingual pupils can be explained by the fact that the majority of the primary schools participating in the French programs were located in big cities with a large percentage of families with a migration background.

**3. Results**

Using ANOVAs, linear regression analyses and repeated measure analyses of variance, statistical analyses were computed using SPSS version 26 (2019). Only complete data sets were used, which included the scores for the two cognitive tests.

The raw scores which the first graders in the French programs obtained in the tests on attention (KT1), nonverbal intelligence (CPM) and on the productive L2 French grammar measures for the free speaking task (FREMODS-MS) are presented in Table 1, which also includes the results from the ANOVAs comparing CLIL vs. FL2.

**Table 1.** One-way ANOVAs for the program by test scores.

| | Max. Points (Norm Values) | CLIL Program (*n* = 113) *M* (*SD*) | FL2 Program (*n* = 73) *M* (*SD*) | Comparison (ANOVAs) |
|---|---|---|---|---|
| CPM | 36 (norm: 24–26) | 26.2 (6.3) | 26.6 (6.0) | $F$ (1, 185) = 0.174, $p$ = 0.677, $d$ = 0.063 |
| KT1 | 48 (norm: 23–32) | 31.5 (7.2) | 32.2 (6.6) | $F$ (1, 185) = 0.382, $p$ = 0.538, $d$ = 0.093 |
| **FREMODS-MS** | **3** (no norm values) | **0.5752 (0.772)** | **1.1096 (0.906)** | **$F$ (1, 185) = 19.300, $p$ < 0.001, $d$ = 0.663** |

Notes: *n* = sample size, *M* = median, *SD* = standard deviation; significance level 0.05.

As Table 1 shows, the ANOVAS neither yielded any significant differences nor any considerable effect sizes (Cohen 1988) between children in the FL2 and CLIL program for the KT1 and CPM test, indicating that the two groups neither differed with regard to their

attention skills nor with regard to their nonverbal cognitive skills. In addition, both groups scored age appropriately (or slightly above the age norm) in both cognitive tests.

Between-group differences were noted for the FREMODS-MS: the students in the FL2 program (0.6 points) achieved significantly better results than the students in the CLIL program (1.1 points), with medium effects. The possible reasons for this unexpected finding will be discussed below.

Additional ANOVAs comparing the mono- and multilingual children in the FL2 and CLIL programs did not reveal any significant differences for language background by program for the cognitive tests (CMP: $F$ (3, 168) = 2.183, $p$ = 0.141; KT1: $F$ (3, 168) = 0.043, $p$ = 0.836), although significant differences were noted for the FREMODS-MS ($F$ (3, 168) = 10.225, $p$ = 0.002). Post-hoc tests (Tukey) indicated significant differences, with FL2 monolinguals outperforming FL2 multilinguals, monolingual FL2 children outperforming monolingual CLIL children and monolingual FL2 children outperforming multilingual CLIL children ($p$ < 0.05). No significant differences were noted between monolingual and multilingual children in the CLIL or in the FL2 program ($p$ > 0.05).

In order to examine the relationship between the teaching program, nonverbal intelligence, attention and productive L2 French grammar at the end of grade 1, as well as the role of the background factors, namely gender, language background and educational background, we conducted correlation analyses (see Table 2). The educational background was measured by the indication of the mothers' highest educational level.

**Table 2.** Correlation analyses, indicating Pearson correlation and significance values, with significant values in bold typeface.

| (*n* = 186) | FREMODS-MS | Teaching Program | KT 1 | CPM | Gender | Language Background | Educational Background |
|---|---|---|---|---|---|---|---|
| FREMODS-MS | | | | | | | |
| Teaching program | **−0.288 \*\*** | | | | | | |
| KT 1 | **0.187 \*** | −0.029 | | | | | |
| CPM | 0.035 | −0.046 | 0.010 | | | | |
| Gender | −0.041 | −0.006 | −0.103 | 0.043 | | | |
| Language background | 0.005 | 0.036 | −0.039 | −0.106 | −0.102 | | |
| Educational background | −0.062 | 0.032 | −0.071 | 0.076 | 0.062 | −0.129 | |

Notes: *n* = sample size, \* *p* < 0.05, \*\* *p* < 0.01.

Significant correlations were only noted between the teaching program and FREMODS-MS and between KT1 and FREMODS-MS. No significant correlations were, on the other hand, found between the teaching program and the cognitive factors. Neither the students' gender nor their language or educational background correlated significantly with the FREMODS-MS scores, it is thus not likely that they serve as predictors for productive L2 French grammar.

Finally, multiple linear regression analyses (Table 3) were employed to test whether the results of the productive L2 French grammar test, FREMODS-MS, at the end of grade 1 can be predicted by the teaching program (CLIL vs. FL2), as indicated by the correlation analyses and the scores obtained in the attention test KT1.

Model 1 shows that the teaching program (FL2 vs. CLIL) and the cognitive variable attention (KT1) significantly predict the productive L2 French grammar scores, accounting for 11.5% of the variance (see Table 3). In addition to this, Model 2 shows that neither the scores for the CPM nor the students' gender, language background or educational background serve as significant predictors, which was already indicated by the lack of correlation between these variables and the FREMOD-MS grammar scores. Adding these factors even declines the predictive power of the model to 10.4%. An interesting side result is that the predictive power of the teaching program and the attention scores increase when

the factors CPM, language background, educational background and gender are added to the regression analysis (see Table 3, Model 2).

**Table 3.** Multiple linear regressions predicting productive L2 French grammar scores at the end of grade 1.

| | Dependent Variable and Predictors | | | | |
|---|---|---|---|---|---|
| | Productive L2 French Grammar Scores in Grade 1 (*n* = 186) | | | | |
| | Model 1 *B (SE)* | *β* | Model 2 *B (SE)* | *β* | |
| Constant | 0.944 (0.314) | | 0.754 (0.511) | | |
| **Teaching program** [1] | **−0.518 (0.120)** | **−0.299 \*\*** | **−0.546 (0.129)** | **−0.314 \*\*** | |
| **KT1** | **0.021 (0.008)** | **0.174 \*** | **0.021 (0.009)** | **0.175 \*** | |
| CPM | | | 0.006 (0.010) | 0.041 | |
| Gender [2] | | | 0.085 (0.125) | 0.050 | |
| Language background [3] | | | 0.014 (0.130) | 0.008 | |
| Educational background | | | −0.013 (0.055) | −0.018 | |
| *r*² **corr.** | | **0.115 \*\*\*** | | **0.104 \*\*\*** | |

Notes: [1] FL2 = 0, CLIL = 1; [2] female = 0, male = 1; [3] monolingual = 0, multilingual = 1; *B* = unstandardized regression coefficient, *SE* = standard error, *β* = standardized beta coefficient; * $p < 0.05$, ** $p < 0.01$, *** $p < 0.001$.

## 4. Discussion

In this study we examined the effects of cognitive variables, such as nonverbal intelligence and sustained attention, on productive L2 French grammar, produced by 186 first graders who either attended regular (voluntary) French-as-a-subject (FL2) lessons or CLIL programs.

### 4.1. First Graders' Productive L2 French Grammar

In terms of their productive L2 French grammar, we found significant between-group differences with students in the FL2 programs outperforming their peers in the CLIL programs. Taking into consideration that exposure to the target language should be more intensive in the CLIL program (with 20–25% of the CLIL teaching time conducted in French according to the teachers, or more precisely, four to six 45 min lessons per week) than in the FL2 program (with only two 45 min lessons per week), this result for productive L2 French grammar contradicts the findings in most other studies on L2 learning in CLIL contexts, especially regarding L2 English grammar (e.g., Steinlen 2018; Kersten et al. 2021). It may be speculated that the CLIL teaching time conducted in French was not enough, taking into account the fact that in CLIL, French is used as a working language and is only indirectly part of the teaching content. Informal inspections of additional teacher questionnaires indicated that there were no considerable differences between the teachers of the two programs in terms of their competences in French. All the teachers possessed high qualifications in French (at least level C1 for the CEFR) or were even native speakers of French, who had been educated either in Germany or in France as elementary school teachers. Some of them had originally studied French to become a secondary school teacher in French, but they had later received specific training (*Zweitqualifizierung*) to qualify as elementary school teachers. Nevertheless, the CLIL program was new to all of the teachers and, while observing the lessons, we occasionally noticed that the methodological approach followed in the CLIL classes was sometimes rather similar to the one followed in the FL2 program and that typical CLIL methods, such as grammatical scaffolding to foster students' production of oral French, could have been used more frequently, whereas lexical scaffolding was used commonly.

Moreover, according to the answers in the questionnaires, the teachers in both programs think that grammatical skills are not as important as lexical, pronunciation or or-

thographical skills, which may be explained by the teachers' assumption that grammatical accuracy is not the aim of CLIL lessons.

Possible differences pertaining to the students' social background and the social environment for each school offering a CLIL and/or an FL2 program may also have played a role, but, according to the results of the parent questionnaire, there were no significant differences between the average socioeconomic status in each group, but participants in the CLIL group even tended to have a higher average socioeconomic status than the participants in the FL2 group.

In summary, we may conclude that the methodology used in class, as well as the FL exposure time, were probably the main reasons for the regular FL program students' significantly better results in productive L2 French grammar. As Dalton-Puffer (2011) and Genesee (1987) have pointed out, L2 exposure time, communicative demands and the type of native speaker models are essential components for productive L2 grammar acquisition, and these may have played out differently in the two French programs.

Finally, we did not examine different grammatical phenomena of French (such as the French conjugation system, the combination of constituents in French noun phrases) in more detail (cf. Harley 1984). It is reasonable to assume that some phenomena were learned more successfully than others, as was already noted for older L2 French learners (e.g., Côté 2020; De Clercq and Housen 2019). A longitudinal study following children from preschool to the end of their school career would allow valuable insights into how the production of grammatical phenomena may change over time in programs with differing L2 intensity (Uhl and Piske 2023).

*4.2. The Impact of Cognitive Skills on Productive L2 French Grammar*

Correlation and regression analyses yielded significant effects for attention/concentration on productive L2 French grammar, independent of the intensity of the teaching program. As noted previously, sustained attention plays an important role in acquiring L1 grammar (see West et al. 2021), and this effect apparently transfers to early foreign language grammar acquisition as well, independent of the intensity of the teaching program. However, only limited information was available as to how L2 grammar was addressed in the different French classrooms, and the relationship between grammar teaching strategies in the L2 classroom (for example, with respect to awareness raising activities or "focus on form" tasks) and students' sustained attention is not well understood. What we can infer from the teacher questionnaires is that grammar is not judged as important as lexis, pronunciation or even orthography. As Robinson (2017) pointed out, it is impossible to know with any certainty that learners in CLIL classrooms just like learners in regular L2 classrooms, outside or inside these classrooms, do not also focus their attention on grammar, with the full intention to learn it. Furthermore, the results from this study and previous research show that a clear operationalization of CLIL and regular foreign language teaching in the classroom reality is not as easy as it seems, as systematic adherence to a specific method is rather rare. This difficulty was already reflected by studies in the 1960s and 1970s comparing more traditional teaching methods, such as audiolingualism and more recent cognitive approaches (e.g., Smith 1970).

Regarding the students' background, we found similarities across the programs: regression analyses did not yield any significant effects of students' gender, educational background and language background on productive L2 French grammar. This result parallels the findings in other studies, which showed that girls and boys and minority and majority language children do not differ regarding their L2 grammar skills, irrespective of their age or the L2 program (e.g., Kersten et al. 2021; Steinlen et al. 2019, 2010; Steinlen 2013, 2017, 2018). Such non-existent effects were also noted for other L2 skills, e.g., for reading and writing (see Steinlen 2021, for a review).

The present study, in general, did not find any significant effect of nonverbal intelligence (CPM) on productive L2 French grammar, independent of the intensity of the teaching program, as the correlation and regression analyses indicated. This result was

rather surprising, taking into account that the ability to reason and to make inferences are important aspects of both grammar learning and nonverbal intelligence. However, the role of nonverbal intelligence in L2 grammar learning does not seem to be well understood yet, which is why more in-depth studies are needed, ideally comparing different age groups with different levels of L2 proficiency.

*4.3. Bilingual Programs as 'Elitist' Programs?*

As noted earlier, bilingual programs are often characterized as being 'elitist' because students in bilingual programs often outperform their peers in regular programs in cognitive tests. In this study, this was not the case: there were no significant differences between first graders in CLIL and regular L2 French programs with respect to their scores in the cognitive tests for nonverbal intelligence (CPM) and attention (KT1). Both groups obtained age-appropriate values in these tests. Thus, none of these two programs can be described as being more elitist than the other. Similar results were reported by Lambert and Tucker (1972) for first graders' nonverbal intelligence within the Canadian early French immersion context, although Lambert and Tucker's students were randomly assigned to an immersion or to a regular school program, whereas in our sample the school program was the parents' choice guided by recommendations from the school. Finally, it should be noted that our results are not in line with Zaunbauer and Möller (2007), who found significant differences between immersion and non-immersion students in grade 1 with respect to nonverbal intelligence (but not regarding attention/concentration).

**5. Conclusions**

This study of 186 first graders examined the productive L2 French grammar of students either attending a regular French as a foreign language or a bilingual German–French program. The possible influence of cognitive variables, such as sustained attention and nonverbal intelligence, on the development of productive L2 French grammar was also considered. This is the first study to determine such effects for beginner learners in different programs with French as the target language. The results for productive L2 French grammar showed non-significant effects for nonverbal intelligence (CPM) and child-internal variables (gender, educational background and language background), but significant effects regarding the effects of the teaching programs (regular vs. bilingual) and attention (KT1). Especially, the non-significant effect of nonverbal intelligence (CPM) on productive L2 French grammar and the significantly higher results for the students in the regular L2 French program in comparison with the CLIL students were unexpected. We suspect that only minor differences between L2 exposure time and teaching methodology in the two programs, as well as only minor differences between the students in terms of their cognitive skills and their educational and socioeconomic background, were probably responsible for these results. We assume that in the first grade of primary school, CLIL students will only show advantages over students in regular French as a foreign language programs in terms of their productive L2 French grammar skills if the L2 exposure time is considerably higher in CLIL programs than in regular L2 lessons. In order to determine whether positive effects of time of exposure on productive L2 French grammar and correlations between nonverbal intelligence and productive L2 French grammar can be observed in later grades, longitudinal studies examining the CLIL students' linguistic and cognitive development are required. Finally, receptive skills were not taken into consideration in this study. It focused exclusively on the relationship between cognitive skills and L2 productive grammar skills, because productive skills often seem to be neglected in studies examining the very early stages of learning. Nevertheless, it would be interesting to compare the development of productive and receptive skills over the four years of elementary school.

**Author Contributions:** Conceptualization, A.K.S. and P.U.; methodology, A.K.S. and P.U.; software, P.U.; validation, P.U.; formal analysis, P.U.; investigation, P.U.; resources, A.K.S. and P.U.; data curation, P.U.; writing—original draft preparation, P.U. and A.K.S.; writing—review and editing, T.P., A.K.S. and P.U.; visualization, P.U.; supervision, T.P. and A.K.S.; project administration, P.U. and T.P.;

funding acquisition P.U. and T.P. All authors have read and agreed to the published version of the manuscript.

**Funding:** This research was funded by a grant from the *Stiftung Bildungspakt Bayern*.

**Institutional Review Board Statement:** The study was conducted in accordance with the Declaration of Helsinki, and approved by the Institutional Review Board (or Ethics Committee) of the University of Erlangen-Nuremberg (protocol code: 20210212 01; date of approval: 17 February 2021).

**Informed Consent Statement:** Informed consent was obtained from all subjects involved in the study.

**Data Availability Statement:** The data are not publicly available due to an agreement signed with the Bavarian State Ministry of Education and Cultural Affairs.

**Conflicts of Interest:** The authors declare no conflict of interest.

## Notes

[1]    In this article, we use L2 in the context of instructed L2 acquisition: "Instructed second language acquisition (ISLA) is a subfield of second language acquisition (SLA) that investigates any type of second language (L2) learning or acquisition that occurs as a result of the manipulation of the L2 learning context or processes. [...] The defining feature of L2 instruction is that there is an attempt, either by teachers or instructional materials, to guide, facilitate, and manipulate the process of L2 acquisition" (Loewen 2023). We are aware that some authers prefer the term foreign language (FL).

[2]    Content and Language Integrated Learning (CLIL) is defined by Eurydice (2006, p. 8) as: "[...] all types of provision in which a second language (a foreign, regional or minority language and/or another official state language) is used to teach certain subjects in the curriculum other than languages [sic!] lessons themselves".

[3]    PT offers an account of the stages learners pass through when learning to process L2 morpho-syntactic structures. More specifically, it predicts a basic developmental chronology, or 'processability hierarchy', consisting of six hierarchically ranked developmental stages. While the processing mechanisms in the processability hierarchy are claimed to be universal, the resulting developmental schedules (i.e., which grammatical structures arise at each stage) are language specific.

[4]    The level C1 refers to proficient FL users and is described as: "Can understand a wide range of demanding, longer texts, and recognise implicit meaning. Can express him/herself fluently and spontaneously without much obvious searching for expressions. Can use language flexibly and effectively for social, academic and professional purposes. Can produce clear, well-structured, detailed text on complex subjects, showing controlled use of organisational patterns, connectors and cohesive devices" (Council of Europe 2001, p. 24).

[5]    0 = no school-leaving qualification, 1 = diploma at junior high school (Mittelschule/Hauptschule: Hauptschulabschluss), 2 = intermediate school certificate (Realschule: mittlere Reife), 3 = professional school qualification (*Berufschulabschluss*), 6 = entrance certificate for higher education (*Fachhochschulreife/Hochschulreife*).

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
