# Peer review of "Elementary School First Graders’ Acquisition of Productive L2 French Grammar in Regular and CLIL Programs"

_languages, doi:10.3390/languages8020138_

Round 1
Reviewer 1 Report (New Reviewer)
- A brief summary
This study compared children in Germany that either followed a regular school program (with French as FL) or a French CLIL school program (with French sometimes being the language of instruction). The main aim was to examine the potential effect of type of school (regular/bilingual) on oral productive FL French grammar skills. Additionally, the authors wanted to assess potential effects of cognitive factors, specifically nonverbal intelligence and sustained attention. In contrast to what they expected, the pupils in the regular program outperformed those following the CLIL program. Furthermore, attention, but not intelligence, positively affected oral proficiency in FL French.
- General comments
The authors have written a very nice and easy-to-read introduction and background section in which they also clearly outline the aim and the variables of interest.
One more general issue I kept wondering about was whether any receptive skills were also checked and, if not, why not? The authors themselves mention that CLIL students' productive skills don’t always match their receptive skills and that some researchers suggested that mainstream FL learners might be better in their oral skills than CLIL pupils (which indeed appears to be the case in this study). Based on this study, you cannot really say whether it’s the overall proficiency that’s lower for the CLIL group as you only assessed production, which might be the skill that lags behind. It would thus be interesting to see whether their receptive skills do match or even exceed those of the regular FL learners. I know the methods cannot be changed anymore, but maybe the authors can address why receptive skills were not assessed?
The abstract as well as the background refer to the potential effect of cognitive variables on the proficiency level obtained, but this seems missing in the overview of research questions. Shouldn’t the cognitive variables (intelligence and attention) be mentioned in research question 3?
Good to see that the groups were relatively well-matched. As for the results, I did not understand why the correlation analyses were performed. The Table suggests that Pearson correlations were used, but how can you correlate (with) nominal variables (e.g., ‘Gender’ and ‘Language Background’)? What would a correlation with a nominal variable mean? As this part of the analyses seems, at least to me, invalid and also unnecessary, maybe this part can just be deleted? I think the main questions can be answered with ANOVA and/or regression. Having said that, I think the explanation of the regression models could be extended a bit. How were the models built and why are there two models in Table 3?
As the authors themselves also mention, most research to date has focused on English and it is very important that studies start to focus on other languages. Specifically, not a lot of research has been done on young learners of French and especially not in the context where this language is a foreign language. This immediately brings me to a point that the authors might want to consider. They use the terms L2 and FL interchangeably and, although this has been done before, it might be better to stick to one of the two terms. In this case, foreign language (FL) makes more sense since French is generally not spoken in Germany. I also think French as a foreign language has been studied less extensively than French as a second language. Thus, using FL as a term might immediately stress the gap this study fills.
Another term-related issue I’d like to raise concerns a reference to internal variables which I initially thought referred to the cognitive measures used (see e.g., sentence 1 of the introduction). The child-internal variables, however, sometimes also only seem to refer to gender and language/educational background (as in the abstract). I think these terms could be used more clearly throughout.
In the conclusion, the unexpected direction of the effect of program should probably be mentioned.
- Specific comments
Page 6, line 175: this seems to start out in a larger font?
Page 6, line 206: I’m not proficient in French, but the comma in the example seems odd to me.
Page 7, Table 1: Maybe replace ‘p=0.000’ with ‘p<0.001’?
Page 8, line 248: we conducted correlation analyses
Author Response
Please see the attachment

Reviewer 2 Report (New Reviewer)
p. 3, lines 74 and 75.
The definitions in parentheticals are rather reductive, if not plain inaccurate. Working memory is hardly about how learners can remember rules, but rather the capacity top hold information in memory so that it may be processed.
p. 3, line 84
“as a cognitive skill” should be in between commas.
p. 3, line 84
Rephrasing is needed. “relates to manipulating or problem solving of”. Perhaps “manipulation or problem solving of”? The sentence reads rather awkward.
p. 4, lines 87 to 107
The section needs to be reworked for organization and style.
p. 4, line 87
This section is intended as literature review, mentions of the instruments included in the present study should come later.
p. 4, line 91
Author(s) need to elaborate on what the disparate results of the studies they mention mean. What do the significant and non-significant between-group results mean as far as non-verbal intelligence and CLIL?
p. 5, line 101
Given the mention of “discovery” later in the same sentence, it seems that “explicit” should be “explicit inductive”.
p. 4, line 95
This claim is highly debatable. Arguably, the population featured in this study is approaching the task of learning a second language largely through implicit mechanisms, and not on a rule-based manner.
p. 4, line 95
“Thus, a prerequisite for learning grammatical rules includes …”
The sentence is a non-sequitur. Nothing about the (alleged) requirement of rule memorization and application of rules implies reasoning, inferencing or analyzing. The way this is phrased, it could be explicit deductive rules being memorized and applied. That would certainly involve no inferencing or and it is questionable whether it would involve reasoning or analyzing.
p. 5, line 15
They did not find any significant between-group differences
The sentence needs rephrasing and elaboration. What does it
p. 5, line 133
Elaboration as to what is meant by receptive grammar is in order, particularly given that this is a question akin to the one asked in this paper.
p. 5, line 157
Please clarify whether 25-30% of the TOTAL teaching time or of the CLIL teaching time is in French.
The fact that after 4 years of CLIL, the expected proficiency is A1 begs for the question as to how much of the CLIL instruction actually takes place in French. Is the instruction in French in CLIL or does the instruction include some French? This is central to the paper, since we are comparing CLIL to FL groups and in FL groups, in a Western country such as Germany, instruction likely happens largely in the Target Language, per the tenets of the Communicative Language Approach. If instruction in CLIL is only partially in French due to the age and proficiency of the learners, this would explain the superiority of the FL group and it would suggest that CLIL is more advisable once a certain degree of proficiency can be counted on.
p. 6, line 175
The initial words are a larger size font.
p. 6, line 188
Reference needed. The claim that explicitly-oriented approaches are helpful with younger learners seems unwarranted, and it should be substantiated.
p. 7, Section 2.3
Further details are needed on the linguistic background of participants. The high percentages of multilingual speakers should be elaborate on, as knowledge of certain language may make instruction in French or French instruction more accessible. Further, it would be interesting to examine whether any gains recorded could be traced back to the multi or monolingual learners in each group, as it may be suggestive that CLIL’s success is predicated upon a certain . This is more so given the finding that language background correlated with
p. 8, line 15
Literature supporting the notion that mothers’ educational background is in any way predictive of childrens’ educational background is sorely needed here. In absence of appropriate substantiation, such a claim could be read as profoundly inappropriate. Why would this measure exclude fathers’ educational achievement needs to be crystal clear.
Further, elaboration is needed, whether in the main text, a footnote or an appendix, as to how educational background was operationalized, since this measure yields significant results the paper draws conclusions from but the instrument to gauge it is not featured in the Methods section.
pp. 1-5
This is a study about CLIL with early learners. A review of literature of CLIL in general and CLIL with youg learners is conspicuously absent.
p. 7, line 232
“as well as” is not appropriate here, given the negative statements. Please rephrase using “neither/nor”.
p. 9, section 4.1
This being the first point in the discussion section would suggest the point on potential elitism in CLIL is a important one. However, the claim that this reputation clouds CLIL programs in Germany is unsubstantiated. You need to include references next to your claim on p. 3, line 63, or, alternatively, you should reserve discussion of this anecdotal claim to a later point in the discussion, where you offer your results as empirical evidence discrediting a commonly held belief.
p. 10, line 295
Here, as well as in section 2.3, elaboration is needed as to how many schools were involved in the study, and the distribution of participants across those schools. This is important to gauge the effect of intact groups and teacher effects on the results presented here.
Round 2
Reviewer 2 Report (New Reviewer)
footnote 3 has a typo -- consistiong should be consisting.
p. 7, line 286 -- word order needs addressing ("not" should precede "usually")
p. 11, line 467 and thereabouts -- the results of this study are reminiscent of the findings from research in the 60's and 70's. Much like in the Pennsylvania Project (Smith, P. (1970). A comparison of the cognitive and audiolingual approaches to foreign language instruction: Then Pennsylvania foreign language project. Philadelphia: Center for Curriculum Development.), the finding seems to be that operationalizing method (CLIL vs Traditional, in this case) is often next to impossible, as systematic adherence to a particular method of teaching appears to not actually happen all that often. Bringing this up here seems relevant.
Author Response
Please see the attachement

This manuscript is a resubmission of an earlier submission. The following is a list of the peer review reports and author responses from that submission.
Round 1
Reviewer 1 Report
The title (Similar or different? Elementary school children’s acquisition of receptive and productive L2 grammar in grade 1: English and French in regular and bilingual programs) suggests that four groups of German L1 and L2 learners are compared along the same lines and the aim of the study is to test a model on a relatively large sample of young learners of English and French as a foreign language in two types of programs by testing them at the end of their first year at school.
The dependent variables are receptive and productive L2 grammar; however not for both target languages, but one for English and or the other one for French. The study aims to find out to what extent the type of program (FL or CLIL operationalized as amount of exposure in classroom contact in the TL), gender, language background and two cognitive variables (nonverbal intelligence and attention) impact receptive or productive grammar and “the role of grammar instruction in the foreign language classroom”. The last point implies explicit grammar teaching in first grade.
The text is well written, and the main ideas and the data are clearly presented. My critical points concern the overview of the literature, the constructs, the research design, the instruments, and the overall content of the paper, and what its main message may be for early languages in schools.
The literature review is an extensive mixed bag: it includes many publications from a wide range of sources and dates; however, many recent publications on how children of this particular age learn foreign languages, and what previous empirical studies found about the key variables impacting their FL development are missing. Motivation and language learning aptitude are not mentioned, SES is mentioned once and none of these are included as variables. The only outcome measure focuses on grammar, which in my view, is not in line with the aims and curricula of early language teaching programs. Also missing are what the realistic aims and achievement targets of early FL and CLIL programs include, what A1 level of proficiency typically means in certain skills and in grammar in English and French, and how these can be and have been assessed in other studies. The text does include some information on target grammar items, but they seem to be defined in terms of grammatical competence (from line 306) focusing on form rather than meaning formulated in “can do statements”. The authors do not discuss in any detail how well children of this age group tend to be able to perform on the grammatical points specified in French and English (some are typically late acquired in morpheme acquisition sequence studies). Worrying about errors getting fossilized (line 347) may lead to an emphasis on error correction practices and explicit grammar teaching. Both may discourage children and decrease their motivation, induce their anxiety, and harm their self-concept. A more balanced age-appropriate classroom methodology can work well in which follow up activities can help children learn target language structures; however, the approach implicitly promoted in the paper are hardly conducive to children learning languages.
It seems that the authors used tests they had access to, but they failed to take into consideration any model of early FL learning on what the key predictors have been found to be. It is not quite clear what the key constructs are. The authors included multiple definitions and borrowed validated tests of nonverbal intelligence and attention, receptive and productive grammar, but they did not discuss how they validated them with the participants. Also, why these variables were hypothesized to impact grammar in the FLs is not underpinned by the literature review. One would assume that language-related underlying abilities are more strongly related to verbal intelligence than nonverbal intelligence and other components of language learning aptitude, as well as to attitudes, motivation and SES; attention is clearly important in learning all skills and knowledge, not only a FL. In other words, I wonder how the study is meant to contribute to our knowledge about FL and CLIL programs, which practices are most conducive to children’s very early L2 development, and what the implications might be for language policy and classroom practice.
The research design is cross-sectional, children filled in tests, which seemed to take quite a long time, especially knowing how long their attention span was, thus raising ethical concerns. No observations were conducted; no other data were made available on how similarly or differently the programs were implemented. The authors said nothing about piloting and validating their grammar tests.
The journal’s policy relates to two points I found worth pointing out: (1) “Authors should not engage in excessive self-citation of their own work.” If I counted correctly, the authors included over 12 of their previous publications. (2) Materials, including data collection instruments, should be made available. The text does not include any of the instruments (complete with rubrics, length, number and types of items, and how they were scored, etc.); therefore, it is difficult to understand how they worked, and the study is not replicable. It would also be necessary to have some sample performances to see what the children could do in their FLs.
Reviewer 2 Report
This paper reported a study on young learners' grammar acquisition under 4 conditions: French foreign language, French CLIL, English foreign language, and English bilingual immersion.
The paper appears to be grounded in the background literature with sufficient care taken to attempt to account for underlying cognitive abilities as an explanation for grammatical uptake. There is some consistency of the results, which show that students perform better with higher attention capacity.
However, the studies show a number of problems both logically and methodologically.
Methods:
A major problem is the use of two different types of test for the two different contexts. The decision to use a productive test of French and a receptive test of English, as well as the unwillingness (and not as they claim unavailability - this could easily be calculated) on the part of the researchers to provide validity and reliability estimates for the English test, mars any potential generalizability or interpretability in having these studies together. As shown by DeKeyser's (1997) study, receptive and productive skills are different, and the lack of justification for choosing these two different tests is striking. While the authors are careful to note how differences in language and test format may explain the differences in the run-on sentence on page 14 from lines 584-591, this recognition needs to be made in direct, plain language: the most likely reason for the differences in the results is not the difference in program, but the difference in test approach. All other comparisons between the programs need to be removed or cast as speculation.
Second, the comparison of CLIL, IM, and "regular" foreign language classes is vexing. Numerous authors have indicated the differences between CLIL and immersion. Though there is an increase in input using CLIL, this does not guarantee an input in student output, unlike in IM contexts. This shows a further mismatch between the instruments and contexts.
In the regression chart for English, is the coding accurately reported? If EFL is 0 and IM is 1, this seems to reverse the trend indicated in the ANOVA (which should be compatible). It looks as if this coding is reversed - please confirm this is correct, as it seems mathematically unlikely for the IM program to have a higher score but to be negatively predictive in the regression.
While a small thing, I also suggest reporting the IM / EFL / CLIL / FFL consistently in the ANOVA tables. The "regular" EFL / FFL classes are presented in reverse (EFL on the right, FFL on the left) in each tables.